# High and Low Temperatures Differentially Affect Survival, Reproduction, and Gene Transcription in Male and Female Moths of *Spodoptera frugiperda*

**DOI:** 10.3390/insects14120958

**Published:** 2023-12-17

**Authors:** Yi-Dong Tao, Yu Liu, Xiao-Shuang Wan, Jin Xu, Da-Ying Fu, Jun-Zhong Zhang

**Affiliations:** 1Laboratory of Forest Disaster Warning and Control in Yunnan Province, Faculty of Biodiversity Conservation, Southwest Forestry University, Kunming 650224, China; yidongtao2022@163.com (Y.-D.T.); axl514@126.com (D.-Y.F.); 2Yunnan Key Laboratory of Plateau Wetland Conservation, Restoration and Ecological Services, Southwest Forestry University, Kunming 650224, China

**Keywords:** *Spodoptera frugiperda*, temperature stress, survival, reproduction, heat shock proteins, zinc finger proteins

## Abstract

**Simple Summary:**

Investigating the adaptive mechanism of insects to different and ecologically relevant temperatures can help in prediction of abundances and control of pests. Here, we found that both heat and cold stresses significantly affected the survival of eggs, larvae, pupae, and adults as well as the reproductive fitness of both sexes in *Spodoptera frugiperda*. Further, based on differential transcriptome analysis in adults, we found that temperature stresses induced sex-specific transcriptional responses that were related to heat or cold detection and resistance, suggesting that male and female moths may adopt different strategies to cope with heat and cold stresses. The results of this study suggest that the negative effects of cold and heat stresses on survival and reproduction may be the consequence of damage to the insect body and cellular microenvironment caused by thermal stress as well as indirect effects from increased expenditure on anti-stress responses. This study provides new information for understanding the stress responses of moths and contributes to the adaptive prediction and control of this pest.

**Abstract:**

In this study, we found that both heat and cold stresses significantly affected the survival and reproduction of both sexes in *Spodoptera frugiperda* adults, with larvae showing relatively higher extreme temperature tolerance. Further transcriptomic analysis in adults found remarkable differences and similarities between sexes in terms of temperature stress responses. Metabolism-related processes were suppressed in heat stressed females, which did not occur to the same extend in males. Moreover, both heat and cold stress reduced immune activities in both sexes. Heat stress induced the upregulation of many heat shock proteins in both sexes, whereas the response to cold stress was insignificant. More cold tolerance-related genes, such as cuticle proteins, UDP-glucuronosyltransferase, and facilitated trehalose transporter Tret1, were found upregulated in males, whereas most of these genes were downregulated in females. Moreover, a large number of fatty acid-related genes, such as fatty acid synthases and desaturases, were differentially expressed under heat and cold stresses in both sexes. Heat stress in females induced the upregulation of a large number of zinc finger proteins and reproduction-related genes; whereas cold stress induced downregulation in genes linked to reproduction. In addition, TRPA1-like encoding genes (which have functions involved in detecting temperature changes) and sex peptide receptor-like genes were found to be differentially expressed in stressed moths. These results indicate sex-specific heat and cold stress responses and adaptive mechanisms and suggest sex-specific trade-offs between stress-resistant progresses and fundamental metabolic processes as well as between survival and reproduction.

## 1. Introduction

The fall armyworm (*Spodoptera frugiperda*; Lepidoptera: Noctuidae) is a major cross-border migratory pest. It invaded China from Myanmar via Yunnan in January 2019, and had spread to 26 provinces/regions by the end of 2019 [1]. This pest has now successfully overwintered in seven provinces/regions of China, including Yunnan, Guangdong, Hainan, Sichuan, Guangxi, Fujian, and Guizhou, and has shown a northward colonization trend [2]. *S. frugiperda* is a tropical and subtropical insect, whereas most areas of China are located in the northern temperate zone. Therefore, environmental temperature is the main factor restricting its expansion, especially its colonization, in China.

Research on temperature stress tolerance is one of the hot spots in the research field around the impact of global climate change on species diversity and adaptive evolution [3], and is at the forefront of insect ecology and molecular biology [4]. In recent years, high-throughput sequencing has been widely used in insect research, providing large-scale genetic information and broadening our understanding of the underlying mechanisms of temperature stress and adaptation [5,6]. These studies have deeply expanded our knowledge of the physiological and genetic background of adaptation to thermal stress, providing a theoretical basis and practical strategies for the prediction and control of agricultural pests.

Heat stress in insects generally causes the upregulation of heat shock proteins (HSPs) and heat shock transcription factors [7]. The HSP family consists of HSP100, HSP90, HSP70, HSP60, and small HSP (sHSP) [8]. HSPs are molecular chaperones, which assist the folding and translocation of newly synthesized proteins [8] and help organisms against a variety of environmental stresses, such as by coping with stress-induced denaturation of other proteins [9,10]. In addition to HSPs, other processes and genes play roles in heat tolerance. Protein subsets of transient receptor potential (TRP) channels, such as TRPA1, TRPV1, TRPM8, and TRPC5, are primarily expressed in sensory neurons and function as sensors to detect thermal and chemical stimuli [11,12]. In insects, the acute stress response begins with the rapid release of a biogenic amine [13], which seems thermoprotective [14], probably by promoting the release of neuroendocrine factors to mobilize energetic reserves [13]. One of the most harmful effects of heat stress is desiccation [15]. Insects cuticles are partly evolved to prevent water loss and infections [16]. A recent study in the rice leaf-folder has shown that cuticular proteins may function in heat stress tolerance [17]. Temperature can affect the fluidity of cell membranes, with heat stress making membranes more fluid and cold stress stiffening them. It is important that membrane fluidity remains optimal in order to ensure the membrane molecular roles [18]. During heat stress, the biosynthesis of saturated fatty acids and sterols may increase, making membranes more rigid [18]. Moreover, heat stress in honeybees induces the upregulation of genes encoding the antioxidant enzyme ascorbate dismutase and the detoxification enzyme cytochrome P450 [19]. RNA-seq analysis in the lesser mulberry pyralid indicates that the immune and phosphatidylinositol signaling systems have a close relationship with heat tolerance [20]. 

At present, insect cold tolerance research mainly focuses on analysis of small-molecule antifreeze protectors and proteins [21]. Small-molecule antifreeze protectants such as glycerol, trehalose and sugars can increase the content of bound water in the insect body, and directly interact with enzymes and other proteins to improve the cold resistance of the insect body. Recent studies in ants and tobacco armyworms have shown that GPDH (glycerol-3-phosphate dehydrogenase) and GK (glycerol kinase) play essential roles in the glycerol biosynthesis and cold resistance mechanisms [22,23]. In wheat midges, it was found that trehalose levels and trehalose synthase (TPS) expression reached their peak under low temperature conditions, while the expression of trehalose degrading enzymes reversed [24]. Unsaturated fat acid in insects can prevent cell membrane lipids from crystallizing at low temperature. In the tobacco armyworm, it was found that the expression of FDA9, a key enzyme in unsaturated fat acid biosynthesis, was upregulated under low temperature conditions [25]. In the European corn borer, *Ostrinia nubilalis*, overwintering diapause led to a significant increase in the overall fat acid unsaturation [26].

With the development and application of high-tech molecular biology methods, other genes that may function in cold tolerance have been detected, such as antifreeze proteins (AFPs) and aquaporins (AQPs). AFPs are a kind of ice binding proteins with thermal hysteresis activity [27,28], while AQPs are transmembrane proteins that may be involved in cold tolerance by controlling water and glycerol flow [29]. 

A previous study has revealed transcriptional changes in response to high and low temperature stress in *S. frugiperda* larvae by using differential transcriptome analysis [5]. In the present study, we further studied the effects of high and low temperatures on the survival, reproduction, and gene transcription of male and female adults in *S. frugiperda*. We discuss the evolutionary significance of the adaptive mechanism and differences between sexes, and aim to provide a basic foundation for discovery of novel genes and pathways involved in temperature stress responses in *S. frugiperda* adults. We conducted research on the adaptation of *S. frugiperda* to environmental temperature stress, and propose that this can help in understanding its tolerance potential to extreme environments while providing ideas and theoretical support for the adaptive prediction and control of this pest.

## 2. Materials and Methods

### 2.1. Insects

*Spodoptera frugiperda* larvae (corn strain) were reared on an artificial diet [30] under conditions of 27 ± 1 °C and 60–80% relative humidity with a 14:10 h light:dark photoperiod. To ensure age and virginity, male and female pupae were sexed according to morphological characteristics [31] and subsequently caged separately. Adult eclosion was recorded daily and emerged male and female moths were reared separately using a 10% honey solution. Under the above rearing condition, the duration of the egg, larva, pupa, and adult stages were about 3 d, 14 d (with six instars), 8 d, and 10 d, respectively.

### 2.2. Effect of Temperature Stress on Survival and Reproduction

To test the effect of temperature stress on the survival of *S. frugiperda*, 1 day-old eggs, third instar larvae, 4 d-old pupae and 3 d-old virgin male and female moths were selected and exposed to different temperatures (−10 °C, −5 °C, 0 °C, 5 °C, 10 °C, 27 °C, 35 °C, 40 °C, and 45 °C) for 3 h to obtain measures of their survival percentages (Figure 1). For eggs, to minimize the effects of handling on egg survival, a complete cluster (ca. 90–180 eggs; *S. frugiperda* eggs are laid in clusters) of 1 d-old eggs was used as one replicate and three replicates were set up for each treatment. Three replicates were used for the other life-stages as well, and 20 insects were used as a replicate. After treatments, the eggs were incubated under 27 °C and the eggs that did not hatch 4 d after incubation were recorded as deaths. For insects of other stages, their deaths were determined 24 h after treatments under 27 °C.

Male and female moths that survived the treatments (all −10 °C and −5 °C treated adults died, and as such were not used for reproduction tests) were collected and paired with non-treated females and males in plastic boxes (25 cm long, 15 cm wide, and 8 cm high; a treated female or male was paired with a non-treated male or female, with one pair per box) for mating and oviposition (Figure 1). The mating event was recorded within two days after pairing. Mated treated females (mated with non-treated males) and non-treated females (mated with treated males) were caged individually in boxes. Each box was provided a paper strip (15 × 20 cm) folded in zigzag fashion as an oviposition substratum and a 10% honey solution as food. Eggs laid within two days after mating were collected and incubated in petri dishes (8.5 × 1.5 cm) under the above conditions. The number of hatched eggs (larvae) was recorded 4 d after incubation. Thirty pairs were used for each sample to measure the number of eggs laid, egg hatching percentage, and longevity (n = 30). Ten pairs of each sample were used as a relocate for mating percentage test; thus, three replicates were used for each sample (n = 3).

Differences in survival percentage, mating percentage, number of eggs laid, egg hatching percentage, and longevity were analyzed using ANOVA followed by LSD tests for multiple comparisons. The percentage data were arcsine transformed to normalize the data prior to analysis. The significance level was set to *p* < 0.05 in all analyses. All values are reported as the mean ± SE.

### 2.3. Effect of Temperature Stress on Gene Transcription

#### 2.3.1. Treatments and Sampling

Three-day-old virgin male and female moths were treated at 40 °C (high temperature, HT), 27 °C (control temperature, CT), and 5 °C (low temperature, LT) for 3 h, as above, and their whole bodies were sampled immediately after treatment (Figure 1). Three males or females were combined to form a sample replicate, and three replicates were used for each sample. All samples were frozen in liquid nitrogen immediately after sampling and stored at −80 °C.

#### 2.3.2. Library Preparation and Sequencing

Total RNA was extracted from samples using TRIzol reagent (Invitrogen, Waltham, MA, USA) following the manufacturer’s protocol. RNA concentration and purity were assayed using a spectrophotometer (Implen, Westlake Village, CA, USA) and a Qubit RNA Assay Kit (Life Technologies, Carlsbad, CA, USA). RNA integrity was measured using an RNA Nano 6000 Assay Kit (Agilent, Santa Clara, CA, USA). Sequencing libraries were prepared using the NEBnext Ultra RNA Library Prep Kit for Illumina (New England BioLabs, Ipswich, MA, USA), and index codes were added to attribute sequences for each sample. The prepared libraries were sequenced on the Illumina HiSeq 4000 platform (Illumina, Foster City, CA, USA) to generate 125 bp/150 bp paired end reads.

#### 2.3.3. Quality Control and Assembly

High-quality clean reads were obtained from raw reads by trimming the adapter and reads containing N (undeterminable base), then deleting low-quality reads using fastp software (v0.19.5) (https://github.com/OpenGene/fastp, accessed on 9 August 2021). The Q20, Q30, and GC contents of the clean data were calculated as well. The obtained clean reads were then mapped to the reference genome sequence of *S. frugiperda* (assembly AGI-APGP CSIRO Sfru_2.0) [32] using Hisat2 software (v2.1.0) (https://daehwankimlab.github.io/hisat2/, accessed on 9 August 2021).

#### 2.3.4. Differential Expression Analysis and Enrichment Analysis of Differentially Expressed Genes

Transcripts per million (TPM) was used to determine gene expression levels, and the edgeR R package (v3.0.8) (http://bioconductor.org/packages/stats/bioc/edgeR/, accessed on 9 August 2021) was used to analyze the differential expression between treatments. The significance threshold of the *p*-value over multiple tests was adjusted by the *q*-value [33], and *q* < 0.05 and |log2(foldchange)| > 1 was used as the threshold to judge the significance of gene expression differences.

The GOSeq program (v2.12) [34] was used to implement GO enrichment analysis and KOBAS software (v2.1.1) [35] was used to perform KEGG enrichment analysis of differentially expressed genes (DEGs). GO terms and KEGG pathways with *q* < 0.05 were recognized as significantly enriched in DEGs.

#### 2.3.5. Validation of RNA-Seq Sequencing Data

Real-time quantitative PCR (qPCR) was performed to verify the accuracy of the RNA-seq data. A total of fifteen DEGs were selected for validation, with GAPDH (ID: LOC118271716) as the reference gene; the PCR primers and functional annotations of the DEGs are shown in Appendix A. RNAiso plus (TaKaRa, Beijing, China) was used to isolate total RNA from samples, and a PrimeScript RT reagent Kit (Takara, China) was used to generate cDNA for qPCR. PCR was carried out on a QuantStudio 7 Flex System (Thermo Fisher Scientific, Waltham, MA, USA), with the reaction conditions being as follows: 95 °C for 30 s, followed by 40 cycles of 95 °C for 5 s, 60 °C for 30 s, and dissociation. The specificity of the SYBR Green PCR signal was confirmed via melting curve analysis. PCR efficiencies were measured by the construction of a standard curve for the target genes and reference gene by 5× serial dilution of cDNA. The Common Base Method [36] was used to calculate the relative expression. Each experiment was repeated three times using three independent RNA samples. Significant differences in the expression of target genes between treatments were analyzed by ANOVA, as above.

## 3. Results

### 3.1. Effect of Temperature Stress on Survival and Reproduction

Temperature stress significantly affected the survival of eggs (*F*_8,18_ = 56.174, *p* < 0.001; Figure 2a), larvae (*F*_8,18_ = 142.324, *p* < 0.001; Figure 2b), pupae (*F*_8,18_ = 587.385, *p* < 0.001; Figure 2c), and adults (*F*_17,36_ = 99.376, *p* < 0.001; Figure 2d). Compared to the control (27 °C), the survival percentage of eggs decreased with the increase or decrease of the treating temperature (*p* < 0.05), with the mortality reaching 100% when the temperature was higher than 40 °C or lower than 0 °C. For larvae, cold stress-resultant death happened when the temperature was lower than −5 °C, whereas no death was detected even when the temperature increased to 45 °C. In pupae and adults, 0 °C treatment significantly reduced the survival percentage (*p* < 0.05), and the percentage decreased to zero when the temperature was lower than −5 °C; 35 °C and 40 °C stress did not show significant effect on the survival (*p* > 0.05), whereas 45 °C stress resulted in 100% mortality in pupae (*p* < 0.05) and up to 85% mortality in adults (*p* < 0.05).

Temperature stress significantly affected the mating percentage (*F*_7,16_ = 12.355, *p* < 0.001; Figure 3a), number of eggs laid within two days after mating (*F*_9,290_ = 11.263, *p* < 0.001; Figure 3b), egg hatching percentage (*F*_9,290_ = 38.068, *p* < 0.001; Figure 3c), and longevity (*F*_9,590_ = 58.530, *p* < 0.001; Figure 3d) of treated adults, with the fecundity and longevity decreasing with the increase or decrease of the treating temperature (*p* < 0.05). Compared to the control (27 °C), the 0 °C and 45 °C treatments reduced the mating percentage and number of eggs laid by more than 50% (*p* < 0.05); the 0 °C treatment reduced the egg hatching percentage by more than 80% (*p* < 0.05); and the 45 °C treatment reduced the egg hatching percentage by 90% in treated females mated with non-treated males and by 35% in non-treated females mated with treated males.

### 3.2. RNA Sequencing and Assembly

RNA-seq obtained 42,100,000–60,700,000 clean reads from each of the eighteen sequenced libraries, with Q20 and Q30 being 97.71–98.27% and 93.61–94.76%, respectively (Appendix A). A total of 31,800,000–45,400,000 clean reads from each of the libraries were mapped to the genome of *S. frugiperda*, with the mapped ratios ranging from 73.68% to 82.26%. Principal component analysis (PCA) showed that biological replicates cluster together (Appendix A), and the Pearson’s correlation coefficient showed higher correlations between biological replicates and lower correlations between treatments (Appendix A), affirming the reproducibility of RNA-seq and biological replicates. The RNA-seq raw reads were deposited into the NCBI SRA database (BioProject ID: PRJNA1017405).

### 3.3. Overview of Transcriptional Changes and Enrichment Analysis

Compared with controls, heat stress induced 2,186 differentially expressed genes (DEGs) in females (HTF vs. CTF) and 1,276 DEGs in males (HTM vs. CTM), while cold stress induced 642 DEGs in females (LTF vs. CTF) and 589 DEGs in males (LTM vs. CTM) (Appendix A). In the heat stress groups, females and males had 272 DEGs in common, while in the cold stress groups, females and males had 89 common DEGs; only twelve common DEGs were shared by all groups (Figure 4, Appendix A).

All DEGs were enriched to GO terms and KEGG pathways, and a total of 266 terms and 54 pathways were significantly (*q* < 0.05) enriched (Appendix A). Most of these terms (170/266 = 63.9%) belong to Biological Process (BP), and most of these pathways (27/54 = 50.0%) belong to metabolism. For a better understanding and summary, these significantly enriched terms and pathways were grouped into sixteen categories based on their function (Figure 5). In addition, the top twenty enriched terms or pathways of each of the comparison groups are presented (Appendix A) and discussed; for those groups that did not have twenty significantly enriched terms or pathways, non-significant (*q* > 0.05) enriched terms or pathways are included. Based on these analyses, the important sex-specific temperature stress-induced molecular changes are explored and described in detail below.

### 3.4. Transcriptional Changes Induced by Heat Stress in Females

Heat stress induced 1088 upregulated and 1098 downregulated DEGs in females (Appendix A). The upregulated DEGs were significantly enriched to sixteen terms/pathways, with two relating to heat tolerance (two DNA methylation), four to genetic information, two to catalysis activities, and four to cellular components (Figure 5a,e, Appendix A). Within the top twenty terms/pathways (Appendix A), one epigenetic regulation (methyltransferase activity, GO:0008168), two DNA repair (damaged DNA binding, GO:0003684; base excision repair, map03410), one reproduction (estrogen signaling pathway, map04915), one longevity (longevity regulating pathway-multiple species, map04213), and one antioxidant (ascorbate and aldarate metabolism, map00053) terms/pathways were found.

The downregulated DEGs were significantly enriched to 186 terms/pathways, of which 62 were related to metabolism (such as metabolism of nucleotide, carbohydrate, energy, amino acid, and sugar), 11 to heat tolerance, 14 to immunity and defense, 21 to genetic information, 14 to catalysis, 16 to transport, 19 to cellular components, etc. (Figure 5a,e, Appendix A). Relatively more energy metabolism-related terms/pathways were found within the top twenty terms/pathways (Appendix A), such as the purine nucleoside triphosphate biosynthetic process (GO:0009145), ATP biosynthetic process (GO:0006754), fructose and mannose metabolism (map00051), and glycerolipid metabolism (map00561).

Specific DEGs that may directly relate to heat stress response and tolerance and reproduction were identified and are presented in Figure 6a, Appendix A. A total of seventeen HSPs were found, with sixteen upregulated and one downregulated, and a 70 kDa HSP (LOC118280708) showing the highest upregulation (258-fold of the control). Two TRPA1 encoding genes were found to be upregulated. Thirteen ubiquitin–protein ligases and eleven DNA repair-related DEGs were found, and most showed upregulation (9/13 and 10/11, respectively). Twenty-seven female reproduction-related DEGs were identified (19 upregulated and 8 downregulated), and most (12/19) of those that were upregulated were associated with egg development. Moreover, a large number (66) of zinc finger protein DEGs were found, with most (54) of them upregulated and others downregulated. In addition, 32 fatty acid-related, 19 cytochrome P450, 12 ATP synthase, and 13 cuticle protein DEGs were found, with most of them being downregulated (10/19, 12/12, and 12/13, respectively). 

### 3.5. Transcriptional Changes Induced by Heat Stress in Males

Heat stress induced 619 upregulated and 657 downregulated DEGs in males (Appendix A). Upregulated DEGs were significantly enriched in 39 terms/pathways, with five relating to heat tolerance (two heat stress proteins, one isomerase activity, and two heat-resistant metabolism), three to immunity and defense, four to response, three to genetic information, five to metabolism, ten to cellular components, two to reproduction, and one to longevity, etc. (Figure 5b,f, Appendix A). Within the top twenty terms/pathways (Appendix A), some organismal protection and heat tolerance related terms/pathways, such as protein folding (GO:0006457) and heat shock protein binding (GO:0031072), as well as antioxidant- and reproduction-related terms/pathways such as ascorbate and aldarate metabolism (map00053) and estrogen signaling pathway (map04915), were found.

Downregulated DEGs were significantly enriched in 45 terms/pathways, including three heat tolerance, fifteen immunity and defense, nine response, three metabolism, eight cellular component, and one longevity, etc. (Figure 5b,f, Appendix A). Within the top twenty terms/pathways (Appendix A), about half of them related to immunity and defense, such as antibacterial humoral response (GO:0019731) and Toll- and Imd-signaling pathway (map04624). Terms/pathways related to metabolism, lifespan, and reproduction, such as glycolysis/gluconeogenesis (map00010), longevity regulating pathway-multiple species (map04213), and estrogen signaling pathway (map04915).

On specific DEGs level (Figure 6b, Appendix A), 28 HSPs were found, with 26 upregulated and two downregulated. A sHSP (HSP 20.4, LOC118280746), also known as protein lethal(2) essential for life-like, showed the highest upregulation (1284-fold of the control). One TRPA1 encoding gene was found to be downregulated. Twenty-one fatty acid-related DEGs were found, with eight upregulated and thirteen downregulated. Moreover, twelve cytochrome P450 were found, with eight upregulated and four downregulated.

### 3.6. Transcriptional Changes Induced by Cold Stress in Females

Cold stress induced 247 upregulated and 395 downregulated DEGs in females (Appendix A). The upregulated DEGs were significantly enriched in four terms/pathways, with one relating to cold tolerance (pentose and glucuronate interconversions, map00040), one response, and two metabolism (Figure 5c,g, Appendix A). Within the top twenty terms/pathways (Appendix A), one histone H2A acetylation (GO:0043968; facilitate gene transcription), one pentose and glucuronate interconversions (map00040, related to cold tolerance), and one ascorbate and aldarate metabolism (map00053; resistance to oxidative stress) were found.

Downregulated DEGs were significantly enriched in 65 terms/pathways, with six relating to cold tolerance, eleven to immunity and defense, twelve to response, twelve to metabolism, and twelve to reproduction (four egg development, three fertilized egg development, two reproductive processes, and two reproductive structures), etc. (Figure 5c,g, Appendix A). Among the top twenty terms/pathways (Appendix A), relatively 22.5% more reproduction related terms/pathways were found, such as eggshell formation (GO:0030703), chorion-containing eggshell formation (GO. 0007304), and prolactin signaling pathway (map04917). Certain pathways associated with organismal protection and cold tolerance, such as the pentose and glucuronate interconversions (map00040), drug metabolism–cytochrome P450 (map00982), and fructose and mannose metabolism (map00051), were found in the top twenty.

Specific DEGs that may directly relate to cold stress response and tolerance and reproduction were identified, and are presented in Figure 6c and Appendix A. Seven HSP DEGs were found, with one upregulated and six downregulated. Six fatty acid-related DEGs were found, with all being downregulated. Twelve facilitated trehalose transporter Tret1 DEGs were found, with most (11) being downregulated and only one upregulated. Eight cytochrome P450 were found, with five upregulated and three downregulated. Sixty-four female reproduction-related DEGs were identified, with four upregulated and 60 downregulated; most (56/60) of the downregulated DEGs were chorion-class proteins. The number of genes in other groups that may relate to cold tolerance was relatively small.

### 3.7. Transcriptional Changes Induced by Cold Stress in Males

Cold stress induced 340 upregulated and 249 downregulated DEGs in males (Appendix A). The upregulated DEGs were significantly enriched in 35 terms/pathways, with five relating to cold tolerance (two detoxification pathways, one pentose and gluconate pathway, one glucuronosyltransferase pathway, and one antioxidant pathway), five to response, six to metabolism, two to reproduction (one embryonic development, one reproductive hormone), etc. (Figure 5d,h; Appendix A). Among the top twenty terms/pathways (Appendix A), more were related to energy metabolism and protein metabolism, such as regulation of generation of precursor metabolites and energy (GO:0043467) and peptidase regulator activity (GO:0061134); some were related to cold tolerance, such as glucuronosyltransferase activity (GO:0015020), UDP-glycosyltransferase activity (GO:0008194) and pentose and glucuronate interconversions (map00040); and others were related to detoxification and reproduction, such as metabolism of xenobiotics by cytochrome P450 (map00980) and steroid hormone biosynthesis (map00140); a pathway related to circadian rhythm, Circadian rhythm-fly (map04711), was found as well.

Downregulated DEGs were significantly enriched in 24 terms/pathways, with one cold tolerance (environmental stress response), ten immunity and defense (five humoral immunity, three immune processes, and two defense responses), five response (four biostimulus response, and one environmental stimulus response), four metabolism, etc. (Figure 5d,h, Appendix A). Among the top twenty terms/pathways (Appendix A), many (16/40) of them were related to immunity and defense or response, such as peptidoglycan binding (GO:0042834) and defense response to other organism (GO. 0098542). Other pathways related to carbohydrate metabolism and cold tolerance, such as carbohydrate digestion and absorption (map04973), fructose and mannose metabolism (map00051), pentose and glucuronate interconversions (map00040), were found in the top twenty as well.

On the specific gene level (Figure 6d, Appendix A), most cold tolerance-related DEGs showed upregulation; eight facilitated trehalose transporter Tret1 (six upregulated and two downregulated), eight UDP-glucuronosyltransferase (six upregulated and two downregulated), four cytochrome P450 (three upregulated and one downregulated), and eight cuticle protein (seven upregulated and one downregulated) DEGs were found. Fourteen fatty acid-related DEGs were found, with eight being upregulated and six downregulated. The number of genes in the other groups was small.

### 3.8. Validation of RNA-Seq Sequencing Data

A total of fifteen DEGs were selected from the four comparison groups (three from HTM vs. CTM, five from HTF vs. CTF, three from LTM vs. CTM, and four from LTF vs. CTF) for validation of the accuracy of RNA-seq. The results showed that the expression levels of target genes from qPCR (Figure 7) were similar to the results from the RNA-seq analysis (Appendix A), suggesting that the RNA-seq data are reliable.

## 4. Discussion

In the present study, we found that both heat and cold stresses significantly affected the survival of eggs, larvae, pupae, and adults as well as the mating percentage, number of eggs laid, and egg hatching success of adults. Survival test showed that larvae had higher extreme temperature tolerance than the other life stages (Figure 2). Field investigations showed that *S. frugiperda* larvae can overwinter in areas with lower temperatures compared to other stages [2], which needs to be considered in the control of this pest. Heat stress in insects may impact the cell microenvironment, such as cell structure and enzyme activity, as well as the spatial conformation of biological macromolecules [37]. Cold stress, on the other hand, causes a decrease in respiratory rate, reduced aerobic respiration, and damage to the cell membrane system, etc. [38,39]. As a result, non-treated development, growth, and survival will be affected. In insects, appropriate temperature is important for the development and hatching of eggs, and extreme temperatures usually result in lower fecundity and egg hatching percentage [40]. For example, in the grain aphid, heat and cold stresses significantly increased the developmental period and reduced aphid fecundity, resulting in a reduced population growth rate [41]. In the present study, we further found that wild females mated with extreme temperature stressed males showed reduced fecundity and egg hatching percentage. This may be because extreme temperature stress may negatively affect ejaculated sperm number [42] and sperm motility and viability [43].

In the mulberry moth pest *Glyphodes pyloalis*, it was found that 1275 DEGs were upregulated and 1222 DEGs were downregulated after 4 h of exposure to 40 °C (compared with 25 °C) [20]. In the present study, compared to controls (27 °C treated groups), heat stress (40 °C) induced 2186 and 1276 DEGs in female and male adults, respectively, whereas cold stress (5 °C) induced fewer DEGs in female and male adults (642 and 589, respectively). A previous study in *S. frugiperda* larvae showed that heat (40 °C) stress induced more DEGs than cold (4 °C) stress (1248 vs. 199 DEGs) compared with the 25 °C treated group [5]. Trait-specific temperature stress responses have been shown in the seed bug *Nysius groenlandicus*, where high temperature induced markedly different expression patterns whereas cold stress responses were not manifested at the transcriptomic level [44]. With regard to the number of significantly enriched terms/pathways, both heat- and cold stress-induced upregulated DEGs in females were enriched in fewer terms/pathways than downregulated DEGs, whereas in males both heat and cold stress induced upregulated and downregulated DEGs enriched in similar numbers of terms/pathways (Figure 5).

On the functional level, the largest difference between sexes in terms of heat stress-induced terms/pathways was in the ‘metabolism’ category (Figure 5), with females enriched in far more metabolism-related terms/pathways (62 terms/pathways, all enriched in downregulated DEGs) than males (eight terms/pathways, with five enriched in upregulated DEGs and three in downregulated DEGs). A similar trend, although not as remarkable, was found in cold stressed groups, with females enriched in fourteen metabolism-related terms/pathways (twelve enriched in downregulated DEGs and two in upregulated DEGs), whereas males were enriched in ten metabolism-related terms/pathways (six enriched in upregulated DEGs and four in downregulated DEGs). In *S. frugiperda* larvae, the DEG study showed that many metabolic process genes were suppressed at low and high temperatures [5]. Generally, the insect metabolic rate increases with temperature; exceptions include flying or behaviorally thermoregulating insects, in which metabolic rates may remain constant or decrease with increasing temperature [45]. Moreover, high temperatures may cause protein denaturation, leading to a decline in metabolism [46], and a reduction in the metabolic rate can possibly lead to a trade-off between stress resistant progresses and fundamental metabolic processes [7]. Fundamental metabolic processes account for up to 50% of individual energy expenditure, and are correlated with survival and individual fitness [47]. Insects exposed to high temperatures need to use antioxidant defenses such as catalase, superoxide dismutase, glutathione-S-transferase, and ascorbic acid, which act together to reduce cellular damage [10,48]. Therefore, reducing the metabolic rate may be an important strategy to escape stressful conditions such as extreme heat [7]. One main mechanism for modulating metabolic rate is reversible protein phosphorylation, carried out by multiple kinase proteins that can control fuel metabolism by modulating transcriptional and translational factors to suppress certain metabolic loci [49]. Thus, the sex-specific heat stress responses on metabolism may be due to sex-specific trade-offs or sex-specific fundamental metabolic rates. Laboratory tests have shown that *S. frugiperda* females have stronger flight ability than males and are usually more active than males (more activities and flights, which cause them lose scales faster) (Xu J., unpublished data, 2023). 

The biggest similarity between sexes was in the n ‘immunity and defense’ category (Figure 5), where both heat and cold stresses induced quite a number of immunity/defense related terms/pathways, with most of them being enriched in downregulated DEGs. A reduction in immune ability at high temperatures occurs in the tropical butterfly *Bicyclus anynana*, and is particularly remarkable under food stress [50]. In the mealworm, *Tenebrio molitor*, immune response was lower at low temperature [51], while in the diamondback moth, *Plutella xylostella*, immune responses decreased in both female and male larvae under cold stress [52]. These results suggest a resource allocation trade-off between heat/cold tolerance and immunity [7]. Alternatively, this may be due to a decrease in the rate of immune related biochemical reactions and/or the pathogens becoming less active at extreme temperatures [53]. Transcriptome sequencing in *Bombyx mori* revealed that the “longevity regulating pathway-multiple species” pathway was involved in diapause preparation [54], whereas this pathway may contribute to heat tolerance in the larvae of *Monochamus alternatus* [55] and *S. frugiperda* [5]. In the present study, this pathway was enriched in both upregulated and downregulated DEGs of heat stressed male moths.

On the gene level, we found that heat stress significantly induced the upregulation of many HSPs in both sexes, 16 in females and 26 in males. In *G. pyloalis* [20] and the ladybird *Propylaea quatuordecimpunctata* [56], heat stress also resulted in overexpression of HSPs, particularly HSP70s. HSPs are a family of molecular chaperones that are produced by cells in response to stressful conditions [7]. They play multiple roles in insect reproduction, such as gamete protection [57], oocyte maturation [58], and trade-offs between reproduction and survival [59]. HSP70 is the most commonly described HSP in the insect heat response [7]. Here, an HSP70 (LOC118280708) showed the highest upregulation (258-fold of the control) in heat stressed females. In heat stressed males, however, an sHSP (LOC118280746) showed the highest upregulation (1284-fold of the control). *S. frugiperda* is native to tropical and subtropical regions in the Americas [60]. The heat stress-induced large-scale HSP responses and remarkable upregulation may suggest that this moth pest has strong heat tolerance and potential to adapt to global climate change. Cold shock may induce the expression of HSPs in insects, especially some sHSPs. For example, HSP19.5, HSP20.8, and HSP21.7 may play a very important role in the cold tolerance of the American leaf miner [61]. In fruit flies, HSP67 and HSP23 can prevent misfolding or aggregation of damaged proteins, and HSP67 may be involved in the physiological process of fruit flies waking up from cold shock-induced coma, thereby protecting fruit flies from frostbite [62]. In the present study, however, no HSP DEG was found in cold stressed males; seven HSP DEGs were detected in cold stressed females, and most (6) of them were downregulated. The upregulated HSP was HSP40, a sHSP. In cold stressed *P. xylostella*, similarly, no HSP DEGs were detected in females, while four HSP DEGs were found in males, with two being upregulated and two downregulated [52]. In the flea beetle *Agasicles hygrophila*, the expression of TRPA1 in eggs elevated to the peak level at 37.5 °C, then fell back to its preferred temperature (25 °C) level at 42.5 °C [63]. In the present study, we found two TRPA1-like genes upregulated in heat stressed females and only one such gene downregulated in heat stressed males. These results suggest that TRPA1 has stage- and sex-specific roles or mechanisms in detecting thermal stimuli and mediating the corresponding responses.

Moreover, a large number (66) of zinc finger protein DEGs were found in heat stressed females, with most (54) of them being upregulated and others downregulated. Zinc finger proteins (ZNFs) are one of the most abundant groups of proteins, and have a wide range of molecular functions. ZNFs are able to interact with DNA, RNA, PAR (poly-ADP-ribose), and other proteins, and are involved in the regulation of cellular processes such as transcriptional regulation, ubiquitin-mediated protein degradation, DNA repair, cell migration, and numerous other processes [64]. Many ZNFs have been confirmed to be involved in both abiotic and biotic stresses [65,66]. In *Apis cerana*, ZFP41 was upregulated during exposure to oxidative stress, including stress induced by extreme temperature; silencing of ZFP41 led to transcriptional changes in antioxidant genes, and overexpression of AcZFP41 enhanced tolerance to oxidative stress in vivo [67]. These results provide a theoretical basis for further research on the function of ZNFs in stress resistance.

In cold stressed groups, obviously more cold tolerance-related genes were found upregulated in males, such as cuticle proteins, UDP-glucuronosyltransferase, and facilitated trehalose transporter Tret1. However, most of these genes were downregulated in females. Studies in *S. frugiperda* larvae likewise found a cuticular protein gene (larval cuticle protein 1-like) that was upregulated under cold stress [5]. Many other species have been shown to have cold-responsive cuticular protein genes, such as beetles [68], stick insects [69], rice plant hoppers [70], and the seabuckthorn moth pest *Eogystia hippophaecolus* [71], suggesting that changes in the insect cuticle may play an important role in adaptation to low temperature. The difference between males and females may be due to sex-specific trade-offs or sex-specific biological characteristics. Similarly, in *P. xylostella*, cold stress caused the upregulation of more such cold resistance genes in males than in females [52]. In *N. groenlandicus*, however, females responded to stress more quickly and strongly than males [44]. These results imply that males and females may adopt different strategies to cope with cold stress, which warrants further study.

Interestingly, heat stress in females induced the upregulation of quite a number (19) of reproduction-related genes (relatively more chorion proteins were found, including five chorion class A proteins and two chorion class B proteins), whereas cold stress induced the downregulation in far more (60) reproduction-related genes. Most of these reproduction-related DEGs were related to egg development. In insects, the chorion has the essential function of protecting the embryo from external agents during development while allowing gas exchange for respiration [72]. In *B. mori,* an earlier study showed that heat shock treatment to pupae resulted in upregulation of vitellogenin expression in both the ovary and fat body [73], and a recent study further demonstrated that thermal induction induced upregulation of chorion class A and B proteins [74]. The upregulation of these genes indicates their role in reproductive protection upon heat shock. In addition, two sex peptide receptor-like genes were found to be upregulated in heat stressed females. In *Drosophila* females, the sex peptide receptor (SPR) acts to detect male sex peptides (SP) and trigger changes in female behavior [75]. Later studies in *B. mori* have suggested that SPR may function as a prothoracicostatic peptide receptor and play stage-specific roles in regulating ecdysteroidogenesis [76]. A recent study in the Asian gypsy moth further found that SPR is involved in development and stress resistance [77]. These results have increased our understanding of the functions of insect SPRs and aided the identification of new targets for the development of environmentally friendly pesticides.

The ratio of unsaturated fatty acids to saturated fatty acids is crucial for maintaining the function of cell membranes [18,78]. In this study, a large number of fatty acid-related genes, such as fatty acid synthases and desaturases, were differentially expressed under heat and cold stresses in both sexes. However, both fatty acid synthase and desaturase may exhibit upregulation or downregulation under both cold and heat shock conditions, which is consistent with the results found in *P. xylostella* [52]. These results further suggest that membrane fatty acid composition is associated with thermal adaptation [18,78,79], while its thermal-related responses and molecular mechanisms are complex. AFPs and AQPs may have functions in insect cold tolerance [27,80]. However, annotation did not find any such genes in *S. frugiperda* in the present study or in the previous study by Xiao et al. [32].

In conclusion, extreme temperature stress significantly reduced the survival and reproductive success of both sexes in *S. frugiperda*, with larvae presenting relatively higher extreme temperature tolerance. Transcriptomic analysis further found that temperature stresses induced sex-specific transcriptional responses related to heat or cold detection and resistance, suggesting that males and females may use different strategies to cope with heat and cold stresses. Temperature stresses may induce sex-specific trade-offs between stress resistant progresses and fundamental metabolic processes, as well as between survival and reproduction. These results imply that the reduction in survival and reproduction caused by thermal stress may be due to heat- or cold-induced direct damage to the cell membrane, enzyme activity, and respiratory rate, as well as indirect effects from increased expenditure on stress-resistant process. These results provide new information on the stress responses of moths, and contribute to the adaptive prediction and control of this pest. 

## Figures and Tables

**Figure 1 insects-14-00958-f001:**
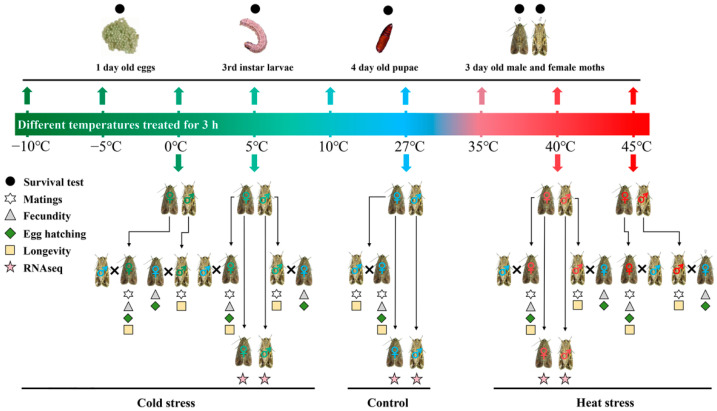
Experimental design. *Spodoptera frugiperda* eggs, larvae, pupae, and virgin male and female moths were treated at different temperatures for the survival test. Surviving male and female moths were paired for reproduction and longevity tests. Virgin male and female moths treated at 40 °C (high temperature, HT), 27 °C (control temperature, CT), and 5 °C (low temperature, LT) for 3 h were sampled for RNA-seq.

**Figure 2 insects-14-00958-f002:**
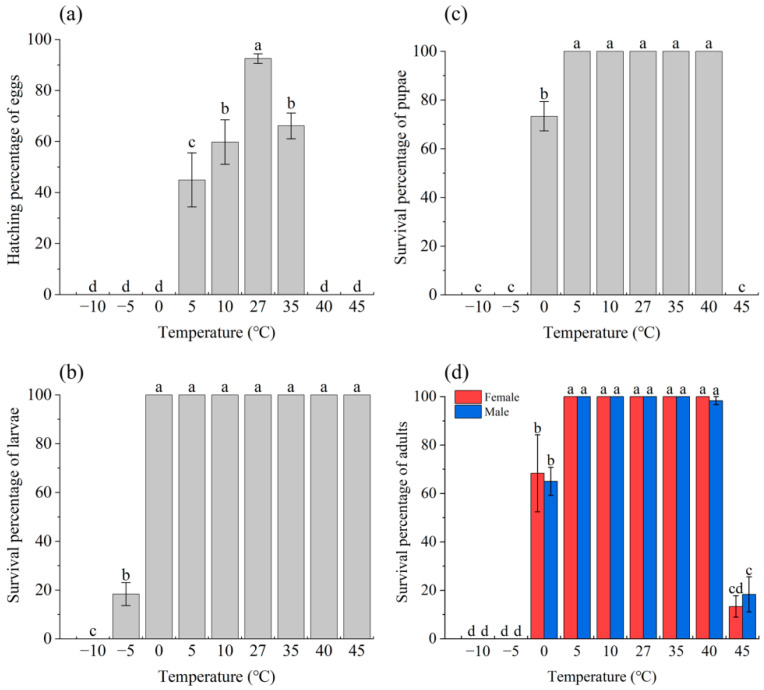
Effect of temperature stress on the survival of *S. frugiperda*: (**a**–**d**) show the results for 1 day-old eggs, third instar larvae, 4 d-old pupae, and 3 d-old adults treated at different temperatures for 3 h, respectively. Three replicates were used for each sample (n = 3). Error bars show the standard error (SE). In each subgraph, bars with different letters are significantly different (*p* < 0.05).

**Figure 3 insects-14-00958-f003:**
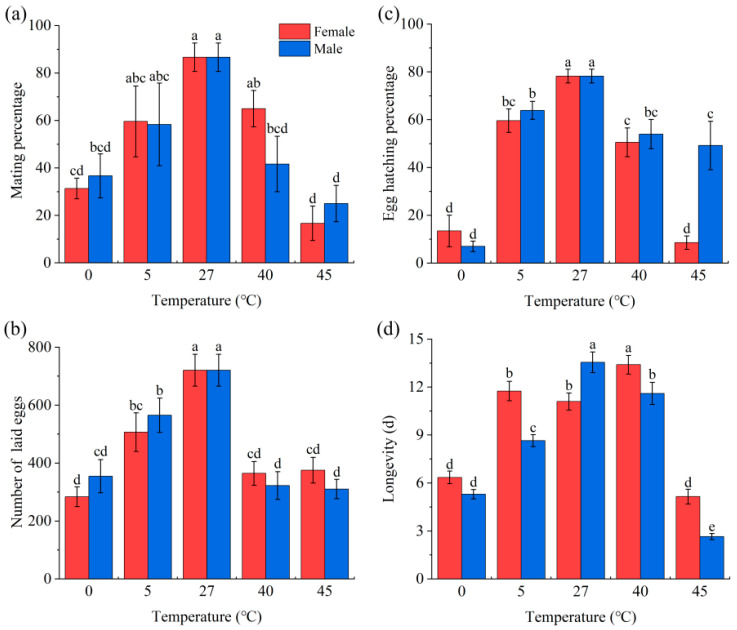
Effect of temperature stress on reproduction and adult longevity in *S. frugiperda*: (**a**–**d**) show the mating percentage, number of eggs laid, egg hatching percentage, and longevity, respectively. Three replicates were used for each sample to measure mating percentage (n = 3), and 30 replicates were used for each sample to measure the number of eggs laid, egg hatching percentage, and longevity (n = 30). Error bars show the standard error (SE). In each subgraph, bars with different letters are significantly different (*p* < 0.05).

**Figure 4 insects-14-00958-f004:**
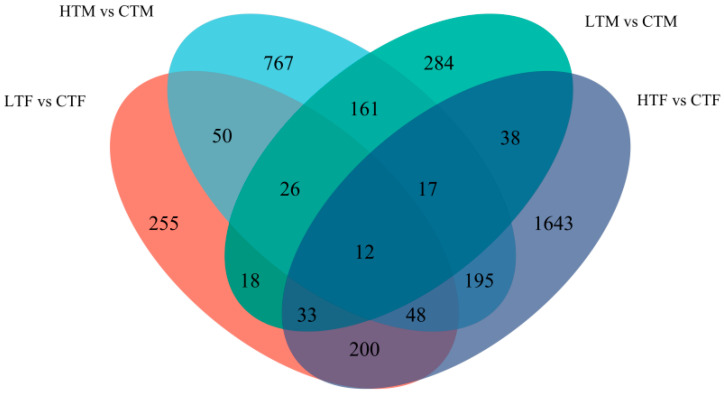
Venn diagram of DEGs in *S. frugiperda*. The overlapping circles represent common DEGs among combinations. HTF, high temperature treated females; HTM, high temperature-treated males; LTF, low temperature-treated female; LTM, low temperature-treated males; CTF, control females; CTM, control males.

**Figure 5 insects-14-00958-f005:**
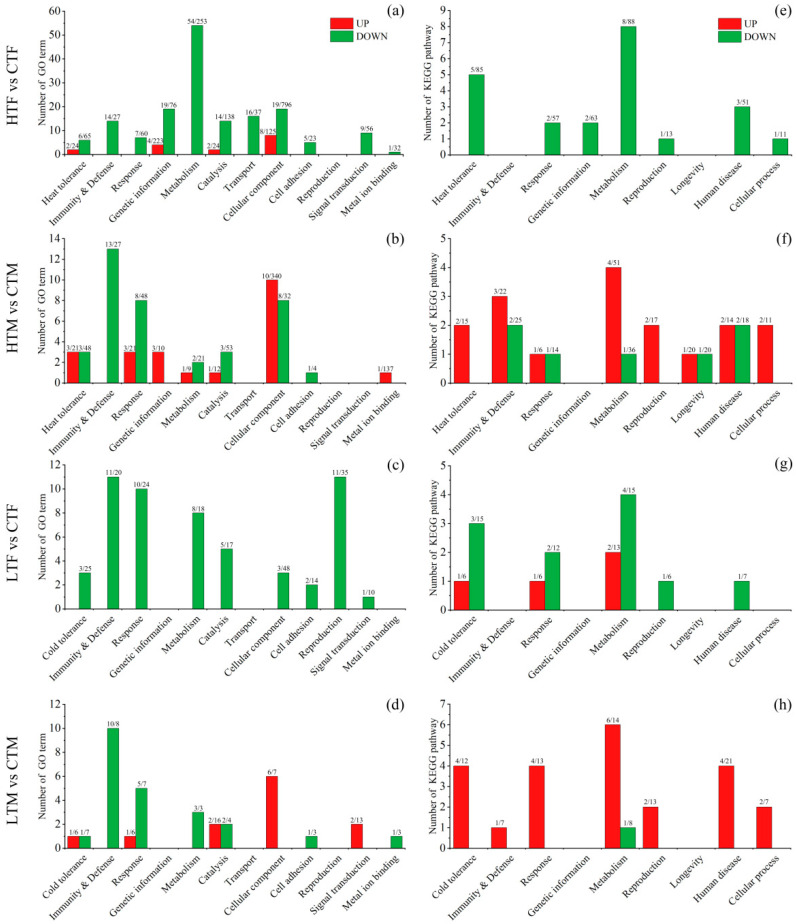
Grouped GO terms and KEGG pathways in *S. frugiperda*: (**a**–**d**) represent GO terms from heat stress females, heat stress males, cold stress females, and cold stress males, respectively; (**e**–**h**) represent KEGG pathways from heat stress females, heat stress males, cold stress females, and cold stress males, respectively. Red columns indicate terms/pathways enriched in upregulated DEGs and green columns indicate terms/pathways enriched in downregulated DEGs. At the top of the column, the first number indicates the number of terms/pathways and the second number indicates the number of DEGs. HTF, high temperature-treated females; HTM, high temperature-treated males; LTF, low temperature-treated females; LTM, low temperature-treated males; CTF, control females; CTM, control males.

**Figure 6 insects-14-00958-f006:**
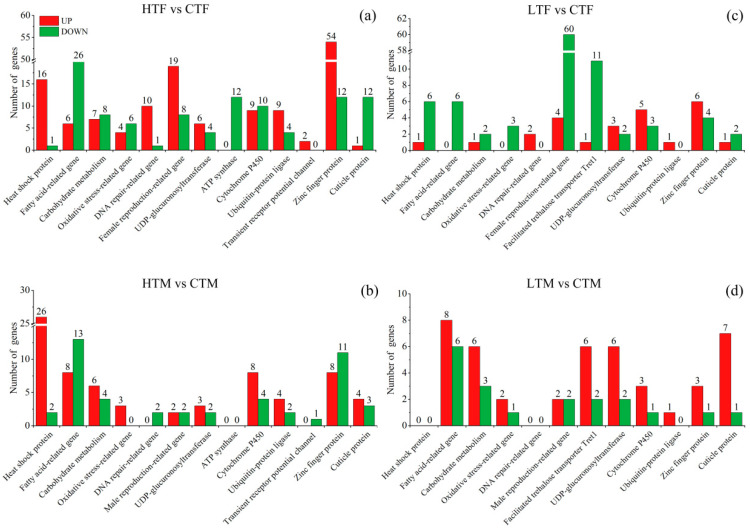
Specific DEGs that may directly relate to temperature stress response and tolerance and reproduction in *S. frugiperda*: (**a**–**d**) are heat stress females, heat stress males, cold stress females, and cold stress males, respectively. HTF, high temperature-treated females; HTM, high temperature-treated males; LTF, low temperature-treated females; LTM, low temperature-treated males; CTF, control females; CTM, control males.

**Figure 7 insects-14-00958-f007:**
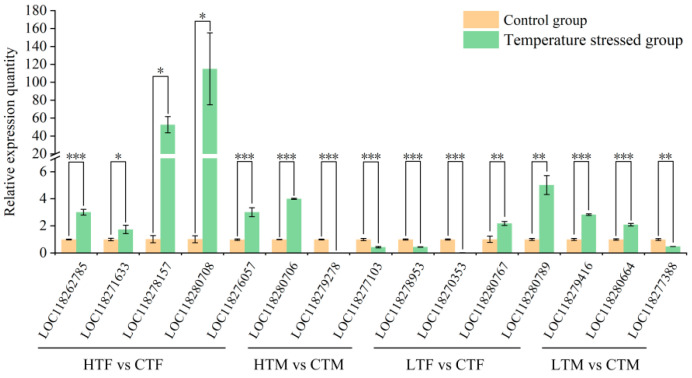
Validation of RNA-seq sequencing data by qPCR in *S. frugiperda.* * *p* < 0.05; ** *p* < 0.01; *** *p* < 0.001. Three replicates were used for each sample (n = 3). The error bars show the standard error (SE). HTF, high temperature-treated females; HTM, high temperature-treated males; LTF, low temperature-treated females; LTM, low temperature-treated males; CTF, control females; CTM, control males.

## Data Availability

The transcriptome raw reads have been deposited in the NCBI SRA database (BioProject ID: PRJNA1017405). Other data generated or analyzed during this study are included in this article and its Appendix A.

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
