# Peer review of "High and Low Temperatures Differentially Affect Survival, Reproduction, and Gene Transcription in Male and Female Moths of Spodoptera frugiperda"

_insects, 2023, doi:10.3390/insects14120958_

Round 1

Reviewer 1 Report

Comments and Suggestions for Authors

In my opinion, the present study on the effects of high and low temperature on survival, reproduction and gene transcription of male and female adults of S. frugiperda is interesting. At the level of content, background and methodology, I have seen the manuscript correct. The results are also well detailed. However, I have some comments on the format and some information that should be included to make the figures more understandable. Consideration should be given to including a discussion to complete the paper. The rest of the comments are included in the attached document.

Comments on the Quality of English Language

Only minor editing required.

Author Response

Comments on the Quality of English Language: Only minor editing required. In my opinion, the present study on the effects of high and low temperature on survival, reproduction and gene transcription of male and female adults of S. frugiperda is interesting. At the level of content, background and methodology, I have seen the manuscript correct. The results are also well detailed. However, I have some comments on the format and some information that should be included to make the figures more understandable. Consideration should be given to including a discussion to complete the paper. The rest of the comments are included in the attached document.

Our answer: We thank you the constructive comments to our MS. We have now revised the paper carefully accordingly. Please see our responses to the comments in the PDF file.

Reviewer 2 Report

Comments and Suggestions for Authors

The comments are provided in a separate pdf file.

Comments on the Quality of English Language

Extensive English language editing needed.

The text is not incomprehensible, but many sentences are confusing, too long and too structured. Sometime they are complex to such extent that reader loses the main idea that originated in the beginning of the sentence.

Moreover some of phrases are inappropriate and some of the phrases are inconsistently used throughout the manuscript.

Author Response

Please see my Word file.

Reviewer 3 Report

Comments and Suggestions for Authors

This is an interesting and comprehensive study investigating heat and cold tolerances and transcriptomic responses to heat and cold temperature stress in a species of moth (Spodoptera frugiperda) that constitutes a rapidly spreading invasive pest species in China.

I am impressed by the amount of work done but don’t think that the authors do a good job in making a coherent study. In its present form the manuscript is highly descriptive, and the results are not discussed sufficiently up against what other people have found. Further I suggest that they relate results back to the background for performing the study, namely what we have learnt that is of relevance for the successful invasion of the species in China. This is currently lacking. In relation to this discussing the strong sex specificities, which I find very interesting, is far from sufficiently covered in the discussion. These shortcomings and the poor English (which I will not comment further on in my review) make me reluctant to recommend this work being published in its current form. In my view data has potential but major efforts are needed to make this into a good paper with impact.

 Suggestions/comments:

Several places the wording ‘trivial’ is used to describe transcriptional responses to cold exposure. It is not clear to me what is meant by this wording, and I believe it is misplaced.

It is stated that male and female pupae were sexed according to morphological characteristics. Can this be done without errors being made?

Authors use ‘survival rate of eggs’, ‘survival rate of larvae’, etc. (e.g. Figure 1). But they do not look at a rate (which is speed of a process) – they obtain estimates of survival percentages. Also, the resolution is not high. Results presented in figure 1 would be more informative if a finer temperature resolution was used in the range close to where animals die from heat or cold stress.

Please use same colours in the different figures.

I suggest moving volcanic plots (Figure 3) to supplemental material

I acknowledge the extra steps taken to validate finding (Figure 10).

The discussion reach more like a resume. Suggest to include much more literature – e.g the rich literature obtained from model species like D. melanogaster on the topic. Try also to me more focussed and discuss the results up against the problem outlined in the Introduction, namely that the species constitute a major pest.

I found the section on metabolic rates very confusing: ‘Therefore reducing metabolic rate may be an important strategy to escape stressful conditions such as extreme heat. One main mechanism for modulating metabolic rate is re-versible protein phosphorylation carried out by multiple kinase proteins, which can control fuel metabolism by modulating transcriptional and translational factors to suppress some metabolic loci…’. Metabolic rate increases with increasing temperatures in ectotherms and it is unclear for me what the take home message from the section is

I suggest ending the discussion with a conclusion stating the main findings and how this relates to the successful spread of the species. Having a one sentence section with information about a specific antifreeze protein is not a good way to end the paper

Comments on the Quality of English Language

Throughout the text the english language should be improved. Many companies are available for that and a native english speaking scientist close to the research field should also proofread the work before possible resubmission.  

Author Response

Please see my Word file.

Round 2

Reviewer 2 Report

Comments and Suggestions for Authors

The current manuscript by Yi-Dong Tao et al. is a revised version of a previously submitted, original manuscript in which the authors first presented the novel data on thermal stress on male and female moths of  Spodoptera frugiperda.

Overall, I find that the manuscript has been greatly improved. I applaud the authors for the amount of work and effort that has gone into addressing all of the earlier comments - like I've already mentioned, the quality has improved significantly.

You included graphical presentation of experimental set up, which I appreciate a lot, so thank you for putting the effort to make it!

1) I have only a minor suggestion I have missed to fill in previous version - I do not find connection between the sentence:

cold stress, on the other hand, causes a decrease in respiratory rate, reduced aerobic respiration, and damage to the cell membrane system, etc. [41]

https://www.ncbi.nlm.nih.gov/pmc/articles/PMC1481651/

In this paper, there is no mention of cold stress at all. So, please change the reference to more appropriate one from insects. Some of references that deal with effects of high and low temperatures on similar lepidopteran species:

https://pubmed.ncbi.nlm.nih.gov/25882225/

https://www.ncbi.nlm.nih.gov/pmc/articles/PMC8647814/

and similar

2) And another a minor/medium suggestion I have is regarding the real time PCR results:

Please update data to Ganger’s method for it would give more precise results. For instance, if the Efficiency is 90%, dCt is 4, one would get following result: 1,9 power 4 equals 13, while if the efficiency is 100% the same results would be: 2 power 4 equals 16. So, the result 13 and 16 differ for 20-25%. Thus, in technical terms, even the 90-110% is the acceptable range for the primer efficiency, this does not mean that variance in efficiency of primers (within the acceptable aforementioned range) does not contribute to difference in gene expression.

Thus, please, reshape the qPCR results and take into consideration to future publication results calculate including efficiencies for each and every primer pair.

Apart from that, I do not have any other remarks and think that your publication could be published as soon as you do this small refinements.

Comments on the Quality of English Language

Check the English language for potential minor errors, misspelling and similar, prior to publishing.

Author Response

Comments and Suggestions for Authors The current manuscript by Yi-Dong Tao et al. is a revised version of a previously submitted, original manuscript in which the authors first presented the novel data on thermal stress on male and female moths of Spodoptera frugiperda. Overall, I find that the manuscript has been greatly improved. I applaud the authors for the amount of work and effort that has gone into addressing all of the earlier comments - like I've already mentioned, the quality has improved significantly. You included graphical presentation of experimental set up, which I appreciate a lot, so thank you for putting the effort to make it! Our answer: We thank the positive comments to our revisions and greatly appreciate your reviewing our MS again. 1) I have only a minor suggestion I have missed to fill in previous version - I do not find connection between the sentence: cold stress, on the other hand, causes a decrease in respiratory rate, reduced aerobic respiration, and damage to the cell membrane system, etc. [41] https://www.ncbi.nlm.nih.gov/pmc/articles/PMC1481651/ In this paper, there is no mention of cold stress at all. So, please change the reference to more appropriate one from insects. Some of references that deal with effects of high and low temperatures on similar lepidopteran species: https://pubmed.ncbi.nlm.nih.gov/25882225/ https://www.ncbi.nlm.nih.gov/pmc/articles/PMC8647814/ and similar Our answer: Thanks and revised the words and references. 2) And another a minor/medium suggestion I have is regarding the real time PCR results: Please update data to Ganger’s method for it would give more precise results. For instance, if the Efficiency is 90%, dCt is 4, one would get following result: 1,9 power 4 equals 13, while if the efficiency is 100% the same results would be: 2 power 4 equals 16. So, the result 13 and 16 differ for 20-25%. Thus, in technical terms, even the 90-110% is the acceptable range for the primer efficiency, this does not mean that variance in efficiency of primers (within the acceptable aforementioned range) does not contribute to difference in gene expression. Thus, please, reshape the qPCR results and take into consideration to future publication results calculate including efficiencies for each and every primer pair. Apart from that, I do not have any other remarks and think that your publication could be published as soon as you do this small refinements. Our answer: We very agree above suggestions and have now revised the qPCR part in Results (Figure 7) and M&M according to Ganger’s method. Comments on the Quality of English Language Check the English language for potential minor errors, misspelling and similar, prior to publishing. Our answer: We agree and have now revised the language very carefully according to them and detailed English corrections by reviewer 3#, which have substantially improved the quality of this paper. We are very grateful for all these contributions to our MS.

Reviewer 3 Report

Comments and Suggestions for Authors

Thank you for improving the ms according to my suggestions. 

I still feel that the language can be improved. Please see suggestions in the attached. 

Comments on the Quality of English Language

Improved but still poor

Author Response

Comments and Suggestions for Authors Thank you for improving the ms according to my suggestions. I still feel that the language can be improved. Please see suggestions in the attached. peer-review-33796969.v2.pdf Comments on the Quality of English Language Improved but still poor Our answer: We thank you for your further comments and corrections to our MS. We have now revised our MS very carefully according to them, which have substantially improved the quality of this paper. We are very grateful for all your contributions to our MS.